# Role of p38 MAPK in Atherosclerosis and Aortic Valve Sclerosis

**DOI:** 10.3390/ijms19123761

**Published:** 2018-11-27

**Authors:** Anna Reustle, Michael Torzewski

**Affiliations:** 1Dr. Margarete-Fischer-Bosch-Institute of Clinical Pharmacology, 70376 Stuttgart, Germany; anna.reustle@ikp-stuttgart.de; 2University of Tuebingen, 72074 Tuebingen, Germany; 3Department of Laboratory Medicine and Hospital Hygiene, Robert Bosch-Hospital, 70376 Stuttgart, Germany

**Keywords:** atherosclerosis, aortic valve sclerosis, aortic valve stenosis, p38, MAPK

## Abstract

Atherosclerosis and aortic valve sclerosis are cardiovascular diseases with an increasing prevalence in western societies. Statins are widely applied in atherosclerosis therapy, whereas no pharmacological interventions are available for the treatment of aortic valve sclerosis. Therefore, valve replacement surgery to prevent acute heart failure is the only option for patients with severe aortic stenosis. Both atherosclerosis and aortic valve sclerosis are not simply the consequence of degenerative processes, but rather diseases driven by inflammatory processes in response to lipid-deposition in the blood vessel wall and the aortic valve, respectively. The p38 mitogen-activated protein kinase (MAPK) is involved in inflammatory signaling and activated in response to various intracellular and extracellular stimuli, including oxidative stress, cytokines, and growth factors, all of which are abundantly present in atherosclerotic and aortic valve sclerotic lesions. The responses generated by p38 MAPK signaling in different cell types present in the lesions are diverse and might support the progression of the diseases. This review summarizes experimental findings relating to p38 MAPK in atherosclerosis and aortic valve sclerosis and discusses potential functions of p38 MAPK in the diseases with the aim of clarifying its eligibility as a pharmacological target.

## 1. Introduction

Cardiovascular diseases are the leading cause of death worldwide [1]. Among the diseases, atherosclerosis is the one with the highest mortality in the western world [2]. Risk factors are associated with western lifestyle and include smoking, hypertension, and high blood glucose, lipid, and cholesterol levels. Atherosclerosis develops in arterial blood vessel walls and most commonly occurs in coronary arteries, in branch points of the carotid artery, and the big leg arteries. The vessel wall consists of three tissue layers: the intima at the luminal side, the media, and the adventitia that is in contact with the surrounding perivascular tissue. The intima is composed of a single layer of endothelial cells and subendothelial connective tissue, providing a barrier between the blood flow and the underlying tissue. The media, mainly composed of vascular smooth muscle cells (SMCs) and elastic connective tissue, is the thickest layer and confers stability and elasticity to the vessel wall. Finally, the adventitia represents the most complex layer, pervaded with nerves and small blood vessels (vaso vasorum) that supply the larger vessel with nerve signals and nutrients to regulate vessel wall function. In atherosclerosis, the diameter of the intima layer increases locally due to proliferation and growth of SMCs, connective tissue deposition, and accumulation of lipids from the blood stream, together forming an atherosclerotic plaque (Figure 1; upper right). As a consequence, the lumen of the vessel narrows, impairing the perfusion of the adjacent tissues. If the plaque is instable and ruptures, the thrombogenic lipid core gets into contact with circulating blood, leading to coagulation and thrombus formation. Patients with atherosclerosis and risk of plaque rupture are treated with statins, which lower blood cholesterol levels and stabilize the plaque, possibly by inhibition of inflammatory processes and modulation of plaque composition [3,4].

Aortic valve stenosis (AVS), not as prevalent as atherosclerosis, is one of the most common indications for cardiac surgery and affects around 12% of the elderly population above the age of 74 [5]. Aortic valve sclerosis, or calcific aortic valve disease (CAVD), is an early stage of AVS and is marked by thickening and calcification of the valve tissue. Common risk factors to develop aortic valve sclerosis are high blood pressure, high blood lipid and cholesterol levels, obesity, diabetes mellitus, smoking, and chronic kidney disease [6,7,8,9,10,11]. The healthy human aortic valve is composed of three thin leaflets (<1 mm), each made up of three layers—the fibrosa on the aortic side, the spongiosa, and the ventricularis on the ventricular side of the valve (Figure 1; upper left). Valve interstitial cells (VICs) are scattered throughout the layers, and endothelial cells line the leaflets on both sides. The layers differ in their extracellular matrix (ECM) composition, providing the leaflets with the stability and flexibility needed to open and close with every contraction of the left ventricle, to allow the blood to enter the aorta and supply the body with oxygen-rich blood. In aortic valve sclerosis, or CAVD, the leaflets are obstructed by fibrosis and calcium deposition, mainly in the fibrosa layer, which impairs their ability to smoothly open and close the passage from the heart to the aorta. As a consequence, the valve narrows and the pressure in the left ventricle rises, leading to increased stress and eventually heart failure. To date, no treatments exist to prevent the development of aortic valve sclerosis, or to halt its progression to AVS. Therefore, valve replacement surgery is the only therapeutic option for patients with severe AVS.

Although atherosclerosis and CAVD are distinct diseases, with differing prevalence and disease manifestations, they share common risk factors such as smoking, obesity, high blood pressure, and elevated blood low density lipoprotein (LDL)-cholesterol levels. Independent of lifestyle choices, individuals may be predisposed to some of these risk factors by genetic mutations. In fact, genome-wide association studies (GWAS) have revealed distinct genomic loci which are frequently affected in individuals with cardiovascular diseases, including genes involved in blood coagulation, inflammation, endothelial cell adhesion, and lipid metabolism and transport [12,13]. The *LPA* gene, encoding the lipoprotein a (Lp(a)), a known cardiovascular risk factor, was also identified in GWAS of aortic valve sclerosis [14,15] and shown to be present in elevated levels in plasma and aortic valves of patients [16,17,18]. The presence of common risk factors and genetic dispositions of atherosclerosis and CAVD highlight the existence of shared disease initiation mechanisms [19]. In both diseases, endothelial damage, followed by lipid insudation and accumulation in the intima or fibrosa layers, respectively, are thought to represent the initiating events. To dispose of excess lipids, macrophages are recruited to the sites by damage-activated endothelial cells. If the lipid burden is too high, macrophages accumulate and transform to lipid-laden foam cells. So called fatty streaks, or intimal xanthoma in the vessel walls are thought to be the signs of such early lesions, although they might as well regress without progression into atherosclerotic plaques [20,21]. During progression however, further immune cells are recruited to the lesions by pro-inflammatory cytokines that are secreted by macrophages, endothelial cells, and lesion smooth muscle cells (SMCs) or VICs. Fibrosis occurs due to cell proliferation and ECM remodeling, leading to thickening of the tissues. The chronic inflammatory environment is thought to furthermore promote the tissue calcification that is seen in both pathologies [22,23,24]. Since immune cell infiltration is an early event and chronic inflammation a suspected driver in both pathologies, therapeutic targeting of inflammatory signaling could represent an instrument to intervene with progression of atherosclerosis as well as aortic valve sclerosis and to avoid the fatal consequences of both diseases.

In the context of chronic inflammation, the p38 mitogen-activated protein kinase (MAPK) pathway has gained attention in the field of both atherosclerosis and CAVD research. p38 MAPK signaling is implicated in diverse biological processes, such as tissue development, cell proliferation, apoptosis, inflammation, and cancer (reviewed in [25]). p38 MAPK is activated by various extracellular inducers of inflammation, which are highly abundant in atherosclerotic and CAVD lesions. To illuminate the role of p38 MAPK signaling in atherosclerosis and aortic valve sclerosis, in this review we summarize relevant experimental findings related to p38 MAPK in both pathologies. To acknowledge the tissue complexity of the diseases, we dissected the findings into the different cell types that make up the lesions and influence disease progression. The aim of this review is to give an overview of p38 MAPK signaling in atherosclerosis and aortic valve sclerosis, and to discuss potential therapeutic implications.

## 2. p38 MAPK Signaling

The p38 MAPKs are members of the mitogen-activated serine/threonine kinase family, together with the extracellular signal-regulated kinases (ERKs) and the c-Jun N-terminal kinases (JNKs). The p38 MAPKs are activated in the presence of certain pathogenic stimuli, such as lipopolysaccharides (LPS), by pro-inflammatory cytokines, or when cells experience extracellular stress, such as ultraviolet radiation, heat shock, or hypoxia. Intracellular stress triggered by miss-folded proteins in the endoplasmic reticulum (ER) or DNA damage can also lead to p38 MAPK activation. Common for all extracellular and intracellular inducers of MAPKs is that binding of the associated ligands to their respective receptors sets in motion a cascade of successive phosphorylation events, where MAPK kinase kinases (MAPKKKs/MEKKs) phosphorylate MAPK kinases (MAPKKs/MKKs/MEKs), which in turn phosphorylate and activate MAPKs. MEK3 and MEK6 are the primary MAPK kinases that phosphorylate the p38 MAPKs, whereas different sets of MEKs mainly activate ERKs and JNKs. MAPK-activated protein kinase 2 (MAPKAPK2/MK2) and the heat shock protein 27 (HSP27) are important downstream targets of activated p38 MAPK, functioning to protect cells from heat shock and osmotic stress [26]. A multitude of other p38 MAPK downstream targets are known today, which execute the cellular responses upon p38 MAPK activation. The cell type and cellular context seem to impact the generated response, which can be as diverse as pro-apoptotic, pro-inflammatory, or anti-proliferative. Comprehensive and exhaustive reviews of the p38 MAPK signaling pathway are provided by others, e.g. Cargnello and Roux, or Coulthard et al. [25,27,28].

Since atherosclerosis and aortic valve sclerosis have become recognized as active, inflammation-driven processes, the p38 MAPK has gained attention in this research field. An increasing number of studies investigating p38 MAPK signaling in different cell types associated with these two cardiovascular diseases have been published in recent years. In the following sections, the experimental findings are summarized and subdivided into the cell types that are present in atherosclerotic and CAVD lesions and that are expected to be involved in disease pathogenesis.

## 3. Endothelial Cells

Endothelial cells (ECs) line the luminal surface of blood vessels and both sides of the aortic valve leaflets. The endothelium builds a protective barrier for the underlying tissue and has important functions in regulating the composition of the associated tissue layers, as well as in adhesion and invasion of inflammatory cells. Vascular endothelium is, furthermore, involved in regulation of the vascular tone, by production of nitric oxide (NO) and other vasomodulators. Healthy endothelium acts atheroprotectively by controlling local blood pressure and by inhibition of inflammation and thrombosis. When the endothelium is damaged, for instance by increased shear stress due to high blood pressure, the endothelial barrier breaks and allows the ingress of blood components into the tissue. As a consequence, ECs become activated to initiate repair of the damaged tissue and disposal of intruded cells and molecules. In the process, ECs express adhesion molecules that allow attachment and invasion of inflammatory cells. Atherosclerosis or CAVD develop if early lesions cannot be resolved and progress into chronically inflamed sites. The presence of excess LDL, and its modification in the damaged tissue, seems to be a crucial factor in the initiation of pathologic lesions [29].

Due to its presence in early atherosclerotic and CAVD lesions, native LDL and/or its modification products might represent important inducers of p38 MAPK signaling at early disease stages. Indeed, LDL has been demonstrated to induce p38 MAPK signaling in ECs [30], and various functions have been identified for the p38 kinase, including upregulation of the cell adhesion molecules E-selectin [30] and vascular cell adhesion protein 1 (VCAM-1) [31] and the chemokine monocyte-chemoattractant protein-1 (MCP-1) [32], all involved in pro-inflammatory signaling and local recruitment of immune cells. Lp(a), consisting of LDL covalently bound to apolipoprotein a, is a known risk factor for cardiovascular diseases and has been shown to increase phosphorylation of p38 MAPK and other kinases in human umbilical-vein endothelial cells (HUVEC), inducing cell growth and migration [33]. Other studies showed that p38 MAPK might be involved in EC migration associated with angiogenesis [34,35], which is observed in atherosclerotic and CAVD lesions [36,37]. p38 MAPK has also been shown to be involved in regulation of EC permeability [38]. Interestingly, high density lipoprotein (HDL), which is inversely correlated with the risk of cardiovascular disease development [39], has been shown to inhibit p38 MAPK activity in HUVEC, leading to decreased interleukin (IL)-6 secretion [40]. In contrast, HDL and oxidized HDL (oxHDL) have been shown to activate p38 MAPK signaling in HUVEC, macrophages, and vascular SMCs in other studies [41,42,43,44]. Which of the described functions of p38 MAPK in ECs are associated with atherosclerosis and/or CAVD disease development and progression, or might even represent drivers or inhibitors of the diseases, warrants further investigation.

Both vascular and valve ECs may undergo endothelial-to-mesenchymal transition (EnMT) accompanied by upregulation of alpha smooth muscle actin expression, the acquisition of contractile properties, and the ability to infiltrate the underlying tissue layers. EnMT of vascular and valve ECs is required during development and maintenance of adult tissue homeostasis by replenishing pools of tissue-resident VICs and SMCs. In the pathogenesis of atherosclerosis and CAVD, however, EnMT allows ECs to transition to VICs or SMCs and to further acquire osteoblast-like properties [45]. Osteoblast-like cells are observed in the lesions, and osteogenic processes are thought to drive tissue calcification in advanced disease stages [36].

The initiation events of EC EnMT are not entirely solved to date. However, mechanical strain has been demonstrated to cause EnMT in valve ECs [46]. The strain increases continually with the rise in tissue calcification, representing a self-replenishing process. In the same study, transforming growth factor beta (TGFβ) and Wnt/β-catenin signaling were demonstrated to drive the strain-induced transformation of ECs to VICs on the cellular level. Interestingly, in other studies, the p38 MAPK has also been shown to be activated when vascular ECs are exposed to shear stress, with impact on actin dynamics and cell-cell alignment [47,48,49]. Whether p38 MAPK impacts EnMT of valve and/or vascular ECs has not been investigated so far.

Despite the diverse p38 MAPK functions that point to a pro-inflammatory and generally disease-promoting character of p38 MAPK signaling in ECs, an in vivo study with an ApoE^−/−^ atherogenic mouse model did not identify an impact of EC-specific knockout of p38 MAPK expression on disease progression and outcome [50]. Although in vivo models better account for the complex environment of a disease, their informative value is compromised by differences between organisms species. Especially when inflammatory signaling and immune processes are involved, mouse models might deviate from the human organism [51,52]. The discrepancy between the in vitro and in vivo findings highlights the need for suitable atherosclerosis and CAVD model systems that account for the complex environment and signaling networks of the diseases.

## 4. Smooth Muscle Cells

Vascular smooth muscle cells (SMCs) are the most abundant cell type in the blood vessel wall, and small numbers of SMCs are also present in healthy valve leaflets, although here, the predominant cell type are VICs (see below). SMCs produce matrix proteins, including collagen, providing the tissue with the required stability and flexibility. In atherosclerosis, vascular SMCs are stimulated to grow, proliferate, and migrate and are highly involved in the thickening of the intima layer of the blood vessel wall. SMCs seem to play a dual role in atherosclerosis progression, as they form protective layers around lipid cores, thereby protecting them from luminal stresses that might provoke plaque rupture and, as a consequence, thrombosis. On the other hand, vascular SMCs are drivers of atherosclerotic progression. In advanced atherosclerotic lesions, SMCs have been shown to undergo enhanced apoptosis, leading to plaque destabilization accompanied by an increased risk of plaque rupture [53,54]. In addition, apoptotic SMCs release matrix vesicles, which serve as nucleation sites for calcium crystals and thereby support plaque calcification [55].

The reason for the increased apoptosis of vascular SMCs in atherosclerotic plaques is not entirely understood. Several lines of evidence, however, point to a potential role of LDL and its modification products in the process [56,57,58]. LDL molecules have also been shown to induce the p38 MAPK pathway in SMCs. Treatment of rat vascular SMCs with oxidized LDL (oxLDL) induced p38 MAPK phosphorylation and its nuclear translocation in a pathway that includes G-protein coupled receptors and the phospholipase C [59]. In this study, activation of p38 MAPK led to increased cytotoxicity in vascular SMCs. Pro-calcific and pro-apoptotic effects of p38 MAPK activation in vascular SMCs, as a consequence of oxLDL-induced cellular ceramide levels, have been reported in other studies [60,61]. At this point we want to note that although oxLDL in this review is used as a general term to refer to oxidized LDL particles, several methods of oxLDL generation do exist. Depending on the method, extensively or minimally oxidized LDL particles are generated that differ by the extent and type of phospholipid and protein modifications within the particles [62]. Importantly, extensively and minimally oxidized LDL particles interact with different pattern recognition receptors, inducing distinct or even opposed cellular processes [63,64]. The studies cited above used extensively oxLDL [59,60] and minimally oxLDL [61], respectively. All studies cited in the following sections used extensively oxLDL in their experiments.

Hypertrophy of vascular SMCs is a common feature of atherosclerosis and is one of the underlying mechanisms of intima thickening observed in pre-atherosclerotic lesions [65]. Angiotensin II (Ang II), an inducer of vascular SMC hypertrophy [66], has been shown to activate the p38 kinase by augmenting intracellular oxidative stress [67,68]. In an alternative pathway, Ang II induces p38 MAPK phosphorylation via epidermal growth factor receptor (EGFR) and TGFβ signaling, with effects on peroxisome proliferator-activated receptor gamma (PPARγ) expression and vascular SMC hypertrophy [69,70]. Furthermore, Ang II-mediated stimulation of p38 MAPK signaling has been shown to increase vascular SMCs migration [71] and proliferation [72], all processes involved in the aberrant growth of the intima in atherosclerosis.

As already mentioned above, the role of vascular SMCs in atherosclerosis is dual. In the development of atherosclerotic lesions, proliferation, migration, and cell growth of SMCs seem to be driving factors, whereas later in disease, when the atherosclerotic plaque has already formed, SMC apoptosis gains more importance. The summarized studies show that p38 MAPK signaling is involved in all disease stages by activating different cellular responses that seem to be dependent on the presence of signaling molecules such as modified LDL (mLDL), ceramide, TGFβ, or Ang II, and others that are not discussed in this review. When it comes to LDL, the exact nature of modification and the local concentration further impact the generated response. The matter is actively debated and reviewed elsewhere [29]. In order to dissect the function of p38 MAPK activation in vascular SMCs, more studies are needed. Especially, the activation of p38 kinase in early-to-late atherosclerotic lesions needs to be further assessed, as well as the local context of expression, such as the presence of mLDL and other signaling molecules.

Since SMCs constitute only a minor fraction of cells in the healthy aortic valve, their contribution to CAVD development has not been extensively studied [73,74]. In calcific aortic valves, however, the numbers of SMCs increase and they have been shown to co-localize with calcified regions [75]. The authors of the latter study speculate that in CAVD, TGFβ might be involved in the trans-differentiation of SMCs from cell types present in valve tissues, such as ECs, VICs, or myofibroblasts (MFBs; see below), leading to the observed increase in SMC numbers. Whether p38 MAPK is expressed in valve SMCs, similar to that in vascular SMCs, has not been studied so far.

## 5. Aortic Valve Interstitial Cells, Myofibroblasts, and Vascular Fibroblasts

Valve interstitial cells (VICs) are the dominant cell type in aortic valves. They are found in the fibrosa, spongiosa, and ventricularis layer, and maintain tissue homeostasis by layer-specific ECM deposition. During development and progression of aortic valve sclerosis, VICs take an active part. It has been shown that VICs transform to activated, smooth muscle actin-expressing myofibroblasts (MFBs) in early valve lesions [76]. A subset of MFBs further differentiate into osteoblast-like cells, which express osteogenic factors and produce a calcium-rich bone matrix. It has been furthermore suggested that apoptotic VICs provide initiation sites for calcific nodule formation, additionally supporting aortic valve calcification [77]. Many signaling networks probably interact to drive the pathologic transformation of VICs in aortic valve sclerosis, and so far, the detailed molecular processes and their chronology are not completely understood. The cytokine TGFβ1, however, is thought to play a major role in driving the processes. Through interaction with its cognate receptors expressed on VICs and MFBs, TGFβ1 activates the small mothers against decapentaplegic (SMAD) signaling cascade, which induces the transcription of osteogenic genes, promoting osteogenesis and calcification in the aortic valve [78,79]. In porcine aortic valves, it was shown that TGFβ1 induces calcium nodule formation, generation of reactive oxygen species, and VIC senescence through SMAD, extracellular signal regulated kinase (ERK)1/2, and p38 MAPK signaling [80]. The p38 kinase has also been implicated in TGFβ-mediated induction of the osteogenic transcription factor RUNX2, and seems to be necessary for the osteoblastic differentiation [81,82]. In addition, p38 MAPK was shown to be involved in bone morphogenic protein 2 (BMP2) signaling, which is another potent inducer of SMAD signaling and RUNX2 expression [81,83]. Independent of TGFβ1, oxLDL may induce p38 MAPK as well as JNK phosphorylation in VICs through the pro-osteogenic receptor for advanced glycosylation end-products (RAGE) [84]. In addition, p38 MAPK has been shown to be activated by sphingosine 1-phosphate and LPS in VICs, inducing pro-inflammatory, pro-angiogenic, and osteogenic processes [85]. Therefore, in aortic valve VICs and MFBs, p38 MAPK seems to be involved in major osteogenic signaling pathways and to support the osteogenic transformation of cells that drives tissue calcification.

In atherosclerosis, fibroblasts in the adventitia layer are also activated at early time points and have the ability to transform to MFBs [86]. Whether they contribute to calcification of plaques similar to VICs in aortic valve sclerosis is not known to date. Experimentally, it has been shown that native LDL can induce p38 MAPK signaling in fibroblasts with consequences for cell spreading and morphology [87]. Here, cholesterol was the component of LDL that induced p38 MAPK most potently [88].

## 6. Monocytes and Macrophages

In both atherosclerosis and aortic valve sclerosis, monocytes are recruited to early lesions and differentiate into macrophages once they enter the tissue [89]. Macrophages promote lesion progression in different ways. For one, they support remodeling of the ECM by secretion of proteolytic enzymes such as matrix metalloproteinases (MMPs) and cathepsins that degrade collagens and elastins, providing initiation sites for calcium crystallization [90,91,92]. Secondly, in both pathologies macrophages are transformed to foam cells by the uptake of mLDL. The exact nature of the LDL modification that induces foam cell formation in vivo has not been definitely clarified to date, and different modifications, including oxLDL and enzymatically modified LDL (eLDL), are used to model the process in vitro. Importantly, the uptake of mLDL seems to be required for p38 MAPK activation and foam cell formation in macrophages [93]. Activation of the p38 MAPK pathway has been demonstrated in macrophages associated with atherosclerotic plaques via immunohistochemistry in patient-derived tissues [94] as well as in animal models [95]. Several in vitro studies have investigated the biological consequences of p38 MAPK activation in atherosclerosis-associated macrophages. First of all, p38 MAPK has been demonstrated to be part of a positive feedback mechanism that drives foam cell formation. Here, oxLDL and eLDL induce p38 MAPK activation in macrophages, which in turn enhances LDL uptake by PPARγ-mediated upregulation of LDL uptake receptors such as CD36 [93,94]. In another study, oxLDL has been shown to enhance the adhesive capacity of monocytes by p38 MAPK-mediated upregulation of the chemokine receptor CXCR2 [96]. Furthermore, p38 MAPK activation in macrophages has been shown to induce the expression of pro-inflammatory cytokines in response to mLDL incubation [97,98]. In addition, Senokuchi et al. demonstrated that oxLDL-mediated induction of p38 MAPK is important for the production of the cytokine granulocyte-macrophage colony-stimulating factor (GM-CSF) and proliferation of macrophages [99]. Taken together, the in vitro studies provide concordant evidence for a pro-inflammatory role of p38 MAPK expression in atherosclerosis-associated macrophages, driving disease progression by promotion of macrophage proliferation and chronic inflammation.

In vivo studies in atherosclerotic mouse models on the other hand achieved conflicting results. In a study by Seimon et al. conditional p38 MAPK-deficiency in macrophages of ApoE^-/-^ mice led to increased macrophage apoptosis and atherosclerotic plaque progression induced by ER stress and unfolded protein response [100]. A subsequent study with the same mouse model could not detect any effects of macrophage-specific p38 MAPK depletion on atherosclerotic plaque progression [50]. In an earlier study, systemic depletion of the p38 MAPK downstream kinase MAPK-activated protein kinase-2 (MAPKAPK2) in hypercholesteremic mice resulted in decreased foam cell formation and inflammatory signaling [101].

Similar to atherosclerosis, macrophages infiltrate early valve lesions [76]. To our knowledge, valve-associated macrophages have not been studied so far and therefore their role in disease progression is not clear. In aortic valve sclerosis, lipid-laden macrophages (foam cells) have been detected immunohistologically [76,102], and thus, a similar pro-inflammatory and disease-promoting function as in atherosclerosis might be suspected. Whether p38 MAPK signaling plays a role in valve-associated macrophages and disease progression remains to be investigated.

## 7. Other Immune Cells: Mast Cells, T Cells, Natural Killer T Cells, B Cells, and Dendritic Cells

Besides macrophages, other immune cells are present in atherosclerotic and aortic valve lesions. In the following section, we summarize major findings around other immune cells that are present in atherosclerotic and/or calcified aortic valve lesions, with a focus on p38 MAPK signaling. The immunologic landscape of these lesions is highly diverse and complex, and a detailed description is beyond the scope of this review. Comprehensive reviews that give detailed insight into the relationship of cardiovascular diseases and inflammation have been published in recent years (e.g., [103,104,105,106]).

Mast cells, a leukocyte population that is involved in allergic reactions and wound healing, have been detected in both atherosclerotic and aortic valve lesions [107,108] and shown to induce p38 MAPK phosphorylation in response to oxLDL stimulation in vitro [109]. The authors of the latter study suspected p38 MAPK, together with other MAPKs and NF-κB, to act downstream of toll-like receptor 4 (TLR4), inducing the expression of pro-inflammatory cytokines in the presence of oxLDL, thereby promoting disease progression by recruitment of inflammatory cells and, as a consequence, atherosclerotic plaque destabilization.

T lymphocytes are associated with both calcific nodules in aortic valves [110] and the fibrous cap and plaque in atherosclerosis [111]. The lymphocytes infiltrate early in lesion development, most likely recruited from the blood stream by pro-inflammatory cytokines secreted by macrophages, SMCs and/or VICs. Their entry into the tissue is facilitated by the expression of adhesion molecules, such as VCAM-1, intercellular adhesion molecule 1 (ICAM-1), and P-selectin, by activated endothelial cells. In later disease stages, neo-angiogenesis in the transformed tissue might provide additional access routes for T lymphocytes [112]. T cells are thought to promote lesion progression by maintaining the chronic inflammatory environment that supports tissue remodeling and destabilization. In addition, cytotoxic T cells induce apoptosis in target cells, producing nucleation sites for calcium crystallization [55]. In CAVD, Nagy et al. furthermore showed that activated and clonally expanded cytotoxic CD8^+^ T cells specifically target and kill osteoclasts, a cell type that is usually found in bone tissues and mediates calcium resorption and bone turnover [113]. Although p38 MAPK has been shown to be involved in T cell receptor (TCR) signaling and is important for interferon γ (IFNγ) production in T cells [114,115], to our knowledge, the role of the p38 MAPK signaling pathway in T cells has never been investigated in the context of atherosclerosis or CAVD.

Natural killer T (NKT) cells are a type of T cell that recognizes lipid antigens presented by antigen presenting cells in conjunction with the CD1 surface molecule. In atherosclerosis, NKT cells are present in regions with CD1 expressing foam cells and are suspected to promote the local inflammation by secretion of a variety of cytokines [116]. In agreement with this, NKT cell stimulation exacerbated atherosclerosis in the presence of the CD1 antigen in an ApoE^-/-^ mouse model [117]. The pro-atherogenic function of NKT cells is thought to be a consequence of granzyme B and perforin secretion, two molecules with cytolytic activity [118]. Interestingly, apart from their cytolytic function, NKT cells have been implicated in neo-angiogenesis, which is frequently observed in atherosclerotic plaques and thought to contribute to plaque destabilization [119]. The pro-angiogenic function is thought to be mediated by IL-8 that is secreted by lipid-antigen stimulated NKT cells, which induces epidermal growth factor receptor (EGFR) expression in endothelial cells [120]. NKT cells are also present in sclerotic aortic valves, and have been shown to associate with disease progression in a mouse model [121] and in human valves [122]. The cytokine IL-2 has been shown to upregulate p38 MAPK in NKT cells, resulting in the production of pro- and anti-inflammatory cytokines [123]. Of note, the increased cytokine production mediated by p38 MAPK has been shown to be regulated translationally, but not on the gene expression level [124]. In contrast to these results, in another study, p38 MAPK has been shown to inhibit secretion of IL-2 and IL-4 by NKT cells, and inhibition of p38 MAPK rescued cytokine secretion [125].

B lymphocytes have been detected in atherosclerotic lesions [126,127] and also in calcified heart valves [128]. They gained more attention as antibodies against atherosclerosis-associated epitopes were detected in the lesions and in the blood circulation of patients [129,130]. Generally, two different types of antibodies exist: so called natural antibodies produced by innate-like B-1 cells [131] and antibodies produced by conventional B-2 cells as part of an adaptive immune response. The concept is emerging that natural antibodies targeting autoantigens, including products from oxidative metabolism such as oxLDL, are atheroprotective [132,133], whereas the adaptive immune response mediated by B-2 cells fuels local inflammation and is rather atherogenic [134] (reviewed in [135]). p38 MAPK signaling has not been investigated in the context of atherosclerosis- or CAVD-associated B cells. It has been shown, however, that p38 MAPK is activated upon B cell receptor stimulation, leading to B cell proliferation [136]. The authors of the study moreover showed that p38 MAPK acts in collaboration with the transcription factor MEF2C specifically in B cells that mediate an adaptive immune response. In this regard, p38 MAPK could represent a potential target to reduce atherogenic B cell-mediated inflammatory signaling in atherosclerosis and possibly also in CAVD.

Finally, dendritic cells (DCs) are present in healthy aorta and aortic valves [137]. The endogenous function of DCs is to present antigens to T cells in order to activate an adaptive immune response in the presence of foreign antigens. DCs have also been detected in atherosclerotic lesions [138], especially at sites that are prone to rupture [139]. The role DCs play in atherosclerosis, however, is not well understood. In a review by Koltsova and Ley, studies investigating the impact of DCs for atherosclerosis are summarized, with most studies suggesting a disease-promoting function of DCs, in which they accumulate lipids similar to macrophages and drive local inflammation [140]. No studies investigating the function of DCs in CAVD exist. Experimental findings concerning the importance of the p38 MAPK pathway for DC function are two sided. On the one hand, p38 MAPK activity is required for the maturation of immature DCs [141]. On the other hand, p38 MAPK inhibition in DC progenitor cells leads to enhanced antigen presentation and immune activation [142]. Whether the p38 MAPK pathway is relevant for DC function in atherosclerotic and/or aortic valve lesions, and whether it has an effect on disease development or progression, has not been studied to date.

## 8. Conclusions

Despite the similarities, atherosclerosis and aortic valve sclerosis are distinct in their pathogenesis. This is highlighted by the fact that, although they share common risk factors, not all patients with atherosclerosis are affected by aortic valve sclerosis, and vice versa. In addition, statins that are effectively applied in atherosclerosis therapy show no clinical benefit for patients with aortic valve sclerosis [143,144,145]. Both the blood vessel wall and the aortic valve are complex structures comprised of endothelial cells and underlying connective tissue layers, that are dispersed by ECM-producing SMCs and VICs, respectively. Immune cells are present in healthy tissues, however, their numbers rise dramatically as they are recruited to early lesions and establish sites of chronic inflammation in the courses of the diseases. The p38 MAPK is involved in inflammatory signaling in different settings and cell types and has gained interest in atherosclerosis and CAVD research, especially since these diseases have become recognized as inflammation-driven. With this review we aimed at elucidating the role of p38 MAPK in the development and progression of atherosclerosis and CAVD by outlining its functions in the different cell types that constitute the lesions and impact disease progression. Figure 1 provides an overview of the cell types and the corresponding functions that have been attributed to p38 MAPK activity.

One of the biggest challenges of p38 MAPK research is the multitude of different stimuli that induce its phosphorylation, such as cytokines, growth factors, and osmotic, oxidative, and mechanical stresses (reviewed in [146]), and the difficulty in dissecting the most relevant factors in certain (patho)physiological conditions. In atherosclerosis and CAVD, combinations of such stimulants are present in a spatiotemporal distribution, most likely leading to a variable p38 MAPK activation status throughout lesions. In addition, other cellular pathways, including the c-Jun terminal kinase (JNK), extracellular signal regulated kinase (ERK), and TGFβ signaling crosstalk and interact with components of p38 MAPK signaling, further modulating intracellular signal transduction and eventually the generated response. Finally, p38 MAPK activation leads to different responses in different cell types, which again influence each other, producing dynamic interconnected networks. To date, in vitro experimental models cannot capture the entire complexity of the p38 MAPK signaling network of pathological conditions such as atherosclerosis or CAVD. Nevertheless, they are suitable for the investigation of simplified processes under a controlled environment. Here, well defined experimental conditions and the use of highly specific p38 MAPK inhibitors are requisite for the generation of meaningful results. Studies in animals better account for the complexity of lesions, although certainly none of the models used in atherosclerosis or CAVD research today perfectly resemble the human situation [147]. Therefore, as for in vitro experiments, results obtained with animal models should always be reviewed critically before they are translated to the human organism.

In the end, the question should be answered whether p38 MAPK activation is a driver of human atherosclerosis and/or CAVD or merely a consequence of the pro-inflammatory, stress-laden microenvironment of the lesion. Depending on the outcome, pharmacologic targeting of p38 MAPK with highly specific inhibitors such as skepinone-L [94,148], or targeting of components of the corresponding signaling cascade could become an option for a therapeutic intervention in atherosclerosis and/or CAVD in the future. Indeed, clinical trials with the p38 MAPK inhibitors dilmapimod (SB681323, GalaxoSmithKline, London, UK) and losmapimod (GW856553, GalaxoSmithKline, London, UK) were carried out with atherosclerotic patients and shown to reduce inflammation in atherosclerotic lesions [149,150,151]. However, patients with acute myocardial infarction did not benefit from the treatment with losmapimod in a subsequent phase III trial [152]. Whether patients might benefit from p38 MAPK inhibition at earlier disease stages has, to our knowledge, never been investigated. Especially in the case of CAVD, therapies that slow, halt, or even reverse disease progression are desperately needed to provide an alternative for otherwise inevitable valve replacement procedures.

## Figures and Tables

**Figure 1 ijms-19-03761-f001:**
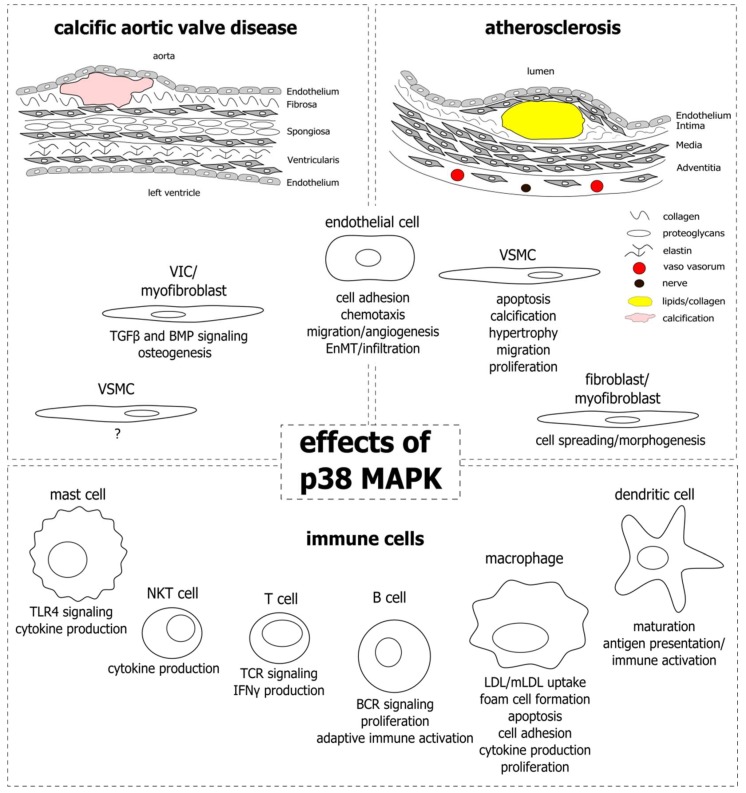
Functional involvement of p38 mitogen-activated protein kinase (MAPK) signaling in calcific aortic valve disease (CAVD) and atherosclerosis. Upper left: CAVD lesion. Schematic cross-section of an aortic valve leaflet composed of the fibrosa, spongiosa, and ventricularis tissue layers. The layers are dispersed by matrix producing valve interstitial cells (VICs) and lined by endothelial cells on both sides which face the aorta or the left ventricle. Lipids accumulate mainly in the collagen-rich fibrosa layer, which is also where the calcification develops. Upper right: atherosclerotic lesion. Schematic cross-section of the vessel wall containing an atherosclerotic plaque. The vessel wall consists of a collagen-rich intima layer lined by endothelial cells that are in direct contact with the blood flow. The underlying media layer contains vascular smooth muscle cells (VSMCs) that contract and dilate in response to nerve signals from the adventitia layer, thereby regulating local blood pressure. The adventitia contains nerves and blood vessels that supply the VSMCs. Atherosclerotic plaques develop in the intima layer and are stabilized by VSMCs from the media. Lower panel: functions attributed to p38 MAPK activity in different immune cells present in CAVD and atherosclerotic lesions. TGFβ: transforming growth factor β; BMP: bone morphogenic protein; EnMT: endothelial to mesenchymal transition; NKT: natural killer T cells; TLR4: toll-like receptor 4; TCR: T cell receptor; IFNγ: interferon γ; BCR: B cell receptor; mLDL: modified LDL.

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
