# Peer review of "Role of p38 MAPK in Atherosclerosis and Aortic Valve Sclerosis"

_ijms, 2018, doi:10.3390/ijms19123761_

Round 1
Reviewer 1 Report
The authors review the role of p38 MAPK in atherosclerosis and aortic valve sclerosis. The Juxtaposition of the two diseases is interesting. However, this reviewer wonders on whether the two diseases share pathomechanisms. THe shared risk factors have different importance for atherosclerosis and aortic valve sclerosis and are confounded by age.
Interestingly, GWAS identified one common genetic risk factor which is not discussed at all by the authors of this review: lipoprotein(a).In this regard it will be interesting to read about the effects of Lp(a) on p38 MAPK. The authors address oxLDL as an inducer of p38 MAPK signaling.Of note there is some overlap between Lp(a) and oxLDL since the majority of oxidized phospholipids is carried by Lp(a). Of note antisense-oligonucleotides lowering Lp(a) by 80-90% is paralleled by lowering of oxLDL levels by the same degree (see work by Tsimikas et al). I therefore suggest that the authors address the role of Lp(a) in the pathogenesis of the two diseases and discuss its effects on p38 MAPK. In this context it is also imporntant to specify what is meant by oxLDL. THere are different procedures of generation and the assays used for measurement of oxLDL differ by their antigens / epitopes. The assay from San Diego / La Jolla measures oxidized phospholipids mainly in Lp(a) whereas the Mercodia assay measures protein modifications. To understand the molecular basis of p38 MAPK activation it will be important to know the specific modification in oxLDL doing this
The authors neither address HDL, which is known to inhibit p38 MAPK at least in endothelial cells .Interestingly, HDL of coronary heart disease patients was reported to have gained a stimulatory effect on p38 MAPK.
Author Response
Response to Reviewer 1 Comments
Point 1: I therefore suggest that the authors address the role of Lp(a) in the pathogenesis of the two diseases and discuss its effects on p38 MAPK.
Response 1: According to the reviewer’s suggestions, we address the role of Lp(a) as a risk factor for the two diseases in paragraph 3 (page 2, lines 70-77). In addition, we searched literature for p38 and Lp(a) and found one study showing that Lp(a) increases p38 MAPK phosphorylation levels in HUVECs. We now cite the study in the “endothelial cell” section (page 5, lines 164-167).
Point 2: To understand the molecular basis of p38 MAPK activation it will be important to know the specific modification in oxLDL doing this
Response 2: In the “smooth muscle cell” section we now point out that different methods of oxLDL generation exist – producing minimally or extensively modified oxLDL (page 6, lines 223-230). All but one of the studies cited in the manuscript use extensively modified oxLDL, which is also noted in the paragraph.
Point 3: The authors neither address HDL, which is known to inhibit p38 MAPK at least in endothelial cells .Interestingly, HDL of coronary heart disease patients was reported to have gained a stimulatory effect on p38 MAPK.
Response 3: HDL has been reported to inhibit and also activate p38 MAPK. We now cite the respective studies (page 5, lines 169-173).
Reviewer 2 Report
This is a well written review that is informative for the reader.
Author Response
Point 1: This is a well written review that is informative for the reader.
Response 1: We thank the reviewer for the positive evaluation of our manuscript.
Reviewer 3 Report
This is a timely review on a highly important player in atherosclerosis and valve calcification: p38 MAPK. Current understanding on this play is poor and somewhat controversial. This manuscript provided an up-to-date review of the research landscape, which will be of high importance and will serve the readership in this field well.
Overall, this is a well-structured and comprehensive review. The organization is clear, and logic flows smoothly. This work demonstrated an excellent view of this field by the authors.
I only have minor comments as follows:
The topic used “aortic valve stenosis (AVS)”. However, calcific aortic valve disease (CAVD), also known as aortic valve sclerosis, seem to be the primary focus of this review, not AVS. AVS is highly heterogeneous diseases and there are many other causes that are out of scope of this manuscript. The author should consider a more accurate title and change languages elsewhere in the text.
Line 56, “up to date” is misuse. It should be “to date”.
The 3rdparagraph should refer to the latest human genetic evidence documenting vascular-wall-specific mechanism contribute to atherosclerosis (PMID: 28714974, 28530674). These were the first comprehensive genetic studies that directly points to vascular wall processes as causal to atherosclerosis. Please consider also briefly state genetic risk factors for arterial calcification (PMID 24098343/ 23561647/ 17903303/ 22144573/ 23870195/ 29221444). Of note, many of these large genetic association studies showed substantial overlapping of risk genetic loci between arterial calcification and myocardial infarction, indicating shared pathological components.
Section 2. p38 MAPK signaling: Please briefly discuss the classification of MAP kinase family members, and unique features and functions that distinguish p38 MAPK from other MAPK family members.
Fig 1 needs careful revision. 1) The background color is confusing. Remove background, or use distinct color schemes for separate topics. 2) What are the big yellow blobs in the vessels? Their legend is missing. If the one on the left is calcification, that should be a different color to distinguish from an atheroplaque on the right. 3) the vessel cross-section of “atherosclerosis” contains dividing lines between intima and media, and between media and adventitia. What are they? The reviewer do not believe those lines are necessary. 4) “capillary” should be vasa vasorum. 5) the lower half shows p38 MAPK surrounded by cells, which makes little sense. What message does the lower half try to convey? If the cells represent different roles of p38 MAPK, change the “p38 MAPK” title to “Roles of p38 MAPK”. Why were “immune cells” labeled under “p38 MAPK”, given that many listed cells (e.g. EC and SMC) are not immune cells? Also, why are there two VSMC? If one VSMC is in valve, label that explicitly as valve SMC.
Author Response
Point 1: The topic used “aortic valve stenosis (AVS)”. However, calcific aortic valve disease (CAVD), also known as aortic valve sclerosis, seem to be the primary focus of this review, not AVS. AVS is highly heterogeneous diseases and there are many other causes that are out of scope of this manuscript. The author should consider a more accurate title and change languages elsewhere in the text.
Response 1: We thank the reviewer for the critical comment. In the title of the manuscript, we use “aortic valve sclerosis” as a synonym for calcific aortic valve disease. We changed the use of “stenosis” to “sclerosis“ in the second paragraph of the introduction (page 2, line 54) and added a short explanatory sentence (page2, lines 52-53), to avoid misunderstandings concerning the topic of the review.
Point 2: Line 56, “up to date” is misuse. It should be “to date”.
Response 2: We thank the reviewer for the comment. The error was corrected (page 2, line 65).
Point 3: The 3rdparagraph should refer to the latest human genetic evidence documenting vascular-wall-specific mechanism contribute to atherosclerosis (PMID: 28714974 , 28530674). These were the first comprehensive genetic studies that directly points to vascular wall processes as causal to atherosclerosis. Please consider also briefly state genetic risk factors for arterial calcification (PMID 24098343/ 23561647/ 17903303/ 22144573/ 23870195/ 29221444). Of note, many of these large genetic association studies showed substantial overlapping of risk genetic loci between arterial calcification and myocardial infarction, indicating shared pathological components.
Response 3: We now mention the genetic risk factors for cardiovascular diseases and aortic valve stenosis in paragraph 3 and refer to the latest GWA studies, as suggested by the reviewer (page 2, lines 70-77).
Point 4: Section 2. p38 MAPK signaling: Please briefly discuss the classification of MAP kinase family members, and unique features and functions that distinguish p38 MAPK from other MAPK family members.
Response 4: We edited the introduction to p38 MAPK signaling according to the reviewer’s suggestions. We now name the other MAPK family members ERK1/2 and JNK and point out important signaling molecules in the p38 MAPK cascade (page 4, lines 119-120 and 128-133).
Point 5: Fig 1 needs careful revision. 1) The background color is confusing. Remove background, or use distinct color schemes for separate topics. 2) What are the big yellow blobs in the vessels? Their legend is missing. If the one on the left is calcification, that should be a different color to distinguish from an atheroplaque on the right. 3) the vessel cross-section of “atherosclerosis” contains dividing lines between intima and media, and between media and adventitia. What are they? The reviewer do not believe those lines are necessary. 4) “capillary” should be vasa vasorum. 5) the lower half shows p38 MAPK surrounded by cells, which makes little sense. What message does the lower half try to convey? If the cells represent different roles of p38 MAPK, change the “p38 MAPK” title to “Roles of p38 MAPK”. Why were “immune cells” labeled under “p38 MAPK”, given that many listed cells (e.g. EC and SMC) are not immune cells? Also, why are there two VSMC? If one VSMC is in valve, label that explicitly as valve SMC.
Response 5: We revised the figure according to the reviewer’s suggestions (page 3). The background colors were removed and boxes were drawn to separate the figure into the fields “calcific aortic valve disease”, “atherosclerosis”, and “immune cells”. The figure and legend was adjusted to discriminate between the atherosclerotic plaque and the calcified valve. The dividing lines between the vessel wall layers were removed. We hope that the adapted figure is clearer and more comprehensible and thank the reviewer for the suggestions.
Reviewer 4 Report
The manuscript by Reustle and Torzewski is an interesting and well-written review on p38 MAPK in atherosclerosis, which also addresses some data in aortic valve disease.
1. As pointed out by the authors at several occasions in the text, little is known about p38 MAPK in aortic valve disease. This questions the relevance of addressing also aortic valve disease in this review.
2. The bibliography is not complete. A quick PubMed search of p38 and valve gives yields several articles, which are not addressed in this study. Here are some examples:
J Heart Valve Dis. 2013 Sep;22(5):621-30.
Arterioscler Thromb Vasc Biol. 2012 Nov;32(11):2711-20
3. The introduction on atherosclerosis is too basic and “textbook style”. Thould be revised to address the pathophysiological mechanisms illustrated in Fig 1.
4. Several important references on risk factors for aortic stenosis are missing. Here are some examples:
Eur Heart J. 2017 Jul 21;38(28):2192-2197
Eur Heart J. 2018 Oct 14;39(39):3596-3603
J Intern Med. 2017 Oct;282(4):332-339
5. I also miss a part on the therapeutic implications of p38 inhibitors, which has been widely addressed in atherosclerosis.
6. The part on smooth muscle cells in the aortic valve is confusing
Author Response
Point 1: As pointed out by the authors at several occasions in the text, little is known about p38 MAPK in aortic valve disease. This questions the relevance of addressing also aortic valve disease in this review.
Response 1: As correctly observed by the reviewer, the literature on p38 MAPK in atherosclerosis is far more extensive than in calcific aortic valve disease (CAVD). We purposefully included CAVD in this review to highlight the need for further research of this condition, especially since no pharmaceutical therapies currently exist. The overlap of CAVD and atherosclerosis in pathophysiology and also in cell types that make up the lesions might allow inferring some of the experimental and clinical findings from atherosclerosis to CAVD, keeping in mind also the differences of the two diseases.
Point 2: The bibliography is not complete. A quick PubMed search of p38 and valve gives yields several articles, which are not addressed in this study.
Response 2: We thank the reviewer for the notice. We expanded our literature research and now include more relevant references in the manuscript, including the ones mentioned by the reviewer (page 7, lines 273-275 and 279-283).
Point 3: The introduction on atherosclerosis is too basic and “textbook style”. Should be revised to address the pathophysiological mechanisms illustrated in Fig 1.
Response 3: We edited the introduction accordingly (page 1-2, lines 34-47). Please note, that the pathological mechanisms are detailed in paragraph 3(page 2, lines 79-89)
Point 4: Several important references on risk factors for aortic stenosis are missing.
Response 4: We added references on the risk factors for aortic stenosis, including the ones suggested by the reviewer (page 2, lines 52-55).
Point 5: I also miss a part on the therapeutic implications of p38 inhibitors, which has been widely addressed in atherosclerosis.
Response 5: We now refer to clinical trials that investigated p38 inhibitors in patients with atherosclerosis in the conclusion of the manuscript (page 10-11, lines 450-456).
Point 6: The part on smooth muscle cells in the aortic valve is confusing.
Response 6: We rephrased the part on smooth muscle cells in the aortic valve to hopefully be less confusing for the reader (page 6, lines 255-257).
Round 2
Reviewer 1 Report
The authors addressed my previous criticisms well.
Reviewer 3 Report
The revised manuscript has fully addressed my concern. I recommend publication.